

# Modeling the effects of water temperature on the population dynamics of *Galba viatrix* and infection by *Fasciola hepatica*: a two-year survey in Andean Patagonia, Argentina

Paula Soler[1,2], Juan Manuel Gurevitz[3], Juan Manuel Morales[3,4] and Marcela Larroza[1]

[1] Grupo de Salud Animal, Instituto Nacional de Tecnología Agropecuaria (INTA), San Carlos de Bariloche, Río Negro, Argentina
[2] INTA-CONICET, Instituto de Investigaciones Forestales y Agropecuarias Bariloche (IFAB), Río Negro, Argentina
[3] Universidad Nacional del Comahue–CONICET, Instituto de Investigaciones en Biodiversidad y Medioambiente (INIBIOMA), Río Negro, Argentina
[4] School of Biodiversity, One Health & Veterinary Medicine, University of Glasgow, Glasgow, United Kingdom

Corresponding author
Paula Soler, soler.paula@inta.gob.ar

## ABSTRACT

**Background:** The trematode parasite *Fasciola hepatica* (liver fluke) can infect livestock, wild mammals, and humans, generating serious economic losses worldwide. Aquatic or amphibious snails of the Lymnaeidae family are the intermediate host of this parasite. Both snail population dynamics and parasite development are closely associated with temperature, although most field studies have recorded air temperature rather than water temperature. Our aim was to statistically model the population dynamics of lymnaeid snails and their infection by *F. hepatica* under natural environmental conditions in Northwest Andean Patagonia.
**Methods:** For two years, we sampled snails monthly in four bodies of water, while registering water and air temperature hourly, and assessing *F. hepatica* infection in snails. Hierarchical Bayesian modeling allowed us to estimate the functional relationship between water temperature and population growth, the probability of detecting snails, and infection by *F. hepatica*.
**Results:** A total of 1,411 *Galba viatrix* snails were collected, identified, and analyzed for *F. hepatica* infection. All sites showed seasonal variation in the number of snails collected and in water temperature as well as sharp variations in snail counts between surveys adjacent in time. The hierarchical model revealed that water temperature acts, at least, at two different time scales: water temperature at the time of sampling determines snail detection probability, whereas the average water temperature between sampling dates affects lymnaeid population growth. We found maximum *F. hepatica* prevalences in snails of 40% (2/5 and 4/10), followed by 33% (65/197). These are the highest prevalences recorded in *G. viatrix* populations in Argentina to date. Our modeling evidenced that the positive effects of water temperature on infection probability increases with snail size and prevalence on the previous survey, while previous prevalence strongly enhances the effects of snail size.

**Conclusions:** Our results underscore the high temporal and spatial variability in the population of snails and the prevalence of *F. hepatica*, as well as the major impact temperature has on detecting snails. Our models provide quantifications of the effects of water temperature on the population growth of *G. viatrix*, its detection, and infection under natural field conditions. These are crucial steps towards generating mechanistic models of *F. hepatica* transmission that would facilitate the design and simulation of potential interventions based on treatments and on environmental and livestock management, taking into account the specific characteristics of each region.

# INTRODUCTION

Fasciolosis is a plant-borne trematode zoonosis caused, in America, by the liver fluke *Fasciola hepatica* (Trematoda: Digenea) (*Mas-Coma, Bargues & Valero, 2005*; *Carmona & Tort, 2017*). It is a major livestock disease that causes serious economic losses due to mortality, the disposal of parasitized livers in slaughterhouses, reduced production of meat, milk, and wool, and increased expenditures in anthelmintics (*Mas-Coma, Bargues & Valero, 2005*; *Aguilar & Olaechea, 2014*; *Kamaludeen et al., 2019*). Moreover, this disease is of public health concern due to an increasing number of human cases being reported (*Mas-Coma, Bargues & Valero, 2005*; *Mas-Coma, Valero & Bargues, 2009*; *World Health Organization, 2014*; *Motta et al., 2023*).

The eradication of fasciolosis is not a realistic goal; instead, control needs to be aimed at the reduction of the disease (*Claxton et al., 1999*; *Torgerson & Claxton, 1999*; *Mas-Coma, Valero & Bargues, 2019*). The development of an effective integrated control strategy requires a thorough understanding of the epidemiology of *F. hepatica*, including the population dynamics of intermediate hosts, snails of the family Lymnaeidae Gasteropoda: Pulmonata (*Malek, 1985*), and how it relates to environmental factors and the climatic conditions of each region (*Malone et al., 1984*; *Yilma & Malone, 1998*; *Fuentes et al., 1999*; *Rojo Vasquez et al., 1999*; *Kleiman et al., 2007*; *Prepelitchi, 2009*; *Bargues et al., 2016*).

Not all species of lymnaeids are equally susceptible to *F. hepatica* infection (*Bargues et al., 2012*; *Sanabria et al., 2012*), therefore, their specific identification is essential (*Torgerson & Claxton, 1999*; *Alda et al., 2021*; *Soler, Abdala & Larroza, 2023*; *Vazquez et al., 2023*). The identification and systematics of lymnaeids have been controversial (*Hubendick, 1951*; *Remigio & Blair, 1997*; *Correa et al., 2010*). In South America, all lymnaeids have been re-classified into a subgroup under the name *Galba* (*Lymnaea*), according to the International Code of Zoological Nomenclature (*Correa et al., 2010*; *Torres, 2022*). In northern Patagonia, where fasciolosis is endemic, molecular identification of lymnaeid species was conducted to detect cryptic species that may exhibit morphological similarities and to determine their role in the transmission of liver fluke (*Standley et al., 2013*; *Soler, Abdala & Larroza, 2023*). To date, only *G. viatrix* has been found infected with *F. hepatica* within this region (*Rubel et al., 2005*; *Cucher et al., 2006*; *Kleiman et al., 2007*; *Olaechea, 2007*; *Soler, 2018*; *Soler, Abdala & Larroza, 2023*). These

studies, in conjunction with prior findings indicating a substantial prevalence of *F. hepatica* infection in grazing animals within the area (60% in sheep and 70% in cattle *Abdala et al., 2022*), substantiates the assertion that *G. viatrix* serves as the principal intermediate host of *F. hepatica* (*Soler, Abdala & Larroza, 2023*).

To estimate the potential risk of infection in grazing livestock, it is essential to consider *G. viatrix* population dynamics and the prevalence of *F. hepatica* stages within the snail (*Cucher et al., 2006*). The prevalence of *F. hepatica* in its intermediate host ranges from 0.01% to 88.3% (*Malone et al., 1984*; *Vazquez et al., 2023*), exhibiting considerable variability not only among different species but also within the same species across different regions (*Gutiérrez, Hernandez & Sánchez, 2005*; *Sanabria et al., 2012*). In Patagonia, prevalences between 0.9% and 14% have been reported (*Rubel et al., 2005*; *Kleiman et al., 2007*; *Larroza et al., 2014*).

Temperature and moisture not only play crucial roles in the development and survival of snails but also influence the activity and abundance of lymnaeids. These snails can adopt survival strategies such as hibernation and aestivation when temperature becomes limiting (*Boray, 1969*; *Ollerenshaw, 1971*; *Malone, 1995*; *Perera et al., 1995*; *Roberts, 1996*; *Kleiman et al., 2007*; *Bargues et al., 2021*). Temperature affects the development of snails and, therefore, their size, which conditions the development of the parasite (*Ollerenshaw & Rowlands, 1959*; *Kendall & Ollerenshaw, 1963*; *Bargues et al., 2021*). In the Andean valleys of Patagonia, even though precipitation concentrates during winter, suitable habitats for the intermediate host are available between spring and autumn, when temperatures reach their maximum (*Kleiman et al., 2007*; *Olaechea, 2007*). It is widely assumed that the dynamics of *G. viatrix* populations follow annual cycles, with increases during the warm season and decreases during winter (*Prepelitchi et al., 2003*; *Kleiman et al., 2007*).

According to several authors, the critical temperature threshold that triggers snail activity is a monthly average temperature of 10 °C for both *G. truncatula* (*Ollerenshaw, 1971*; *Malone et al., 1984*), and *G. viatrix* (*Claxton et al., 1999*; *Boray, Hutchinson & Love, 2007*). Nevertheless, the exact relationship between population growth and temperature under field conditions has not yet been quantified. Furthermore, most field studies have considered air temperature rather than water temperature (*Kleiman et al., 2007*). Temperature can also affect the probability of finding snails during surveys given they are present (*Claxton et al., 1999*; *Kleiman et al., 2007*; *Bargues et al., 2021*; *Rodriguez Quinteros et al., 2024*), and thus detection probability as a function of temperature should be considered when studying the snail population. The aim of this work is to study the population dynamics of *G. viatrix* snails and their infection by *F. hepatica* under natural environmental conditions in an endemic area of Northwest Andean Patagonia. This knowledge is crucial for better understanding and predicting snail population dynamics, abundance, and, ultimately, for guiding precise monitoring and control strategies (*Gurevitz et al., 2011*; *Larroza et al., 2018*).

## MATERIALS AND METHODS

### Study area

The study was conducted in the Andean valley of the Manso Inferior River (41°35′S; 71°26′W), within the Nahuel Huapi National Park, located in the southwest of Río Negro Province, Argentina (Fig. 1A). The area comprises permanent and temporary freshwater environments. The former are represented by lagoons, rivers, and streams, while the latter are represented by ponds, ditches, and water-saturated flat areas of varying extension, which are known as 'mallines'. Four sites were selected, each located in a different body of water, considered suitable habitats for snails (Fig. 1B). The selection of these sites was based on the presence of slow water currents, muddy soils, and year-round accessibility. These characteristics were essential due to the frequent flooding, impassable paths, and vegetation growth in the study area, which can obstruct access to certain locations. Sites 1–3 corresponded to watercourses with depths ranging from 25 to 50 cm. Site 4 was 7–10 cm deep and was connected to a small stream (Fig. 1C).

### Snail survey

We conducted monthly searches for lymnaeid snails from October 2019 to September 2021. During each sampling month across both years, the four selected sites were consistently sampled at the same time of day, between 11 a.m. and 3 p.m. During the colder winter months (June to August), it was presumed that the snails aestivate (*Boray, 1969*; *Olaechea, 2007*; *Kleiman et al., 2007*). To verify this phenomenon at our study site, additional sampling was conducted in July.

A sampling event consisted of collecting all snails found in a 1-m$^2$ quadrat while searching for 30 min (*Relf et al., 2011*) employing a metal scoop with pores 1-mm wide (*Rabinovich, 1980*; *Prepelitchi, 2009*). At each of the four selected sites, 2–3 quadrats were delimited for snail sampling and the geographical coordinates recorded. The number of quadrats selected per site was based on the size of the water body. The quadrats were randomly selected and marked on the GPS to avoid repetition, while encompassing the diversity within the water body (at the shore, in the center, or in between), depending on the characteristics of the water body. In subsequent samplings, different quadrats within each site were selected, separated by a minimum distance of 3 m, to avoid bias from previous searches and ensure accurate sampling of the lymnaeids (Fig. S1). Collected snails were placed alive in labeled plastic containers and kept cool until returning to the laboratory within 24 h of collection. All the surveys were done by the same individual (Paula Soler).

### Morphology and sizing

Snails were examined under a stereoscopic magnifying glass (×40) for morphological identification based on shell shape (*Castellanos & Landoni, 1981*; *Paraense, 1982*; *Samadi et al., 2000*). Lymnaeids were distinguished from other snails by their characteristic smooth, dextral conical shells and their pair of triangular tentacles. Measurements were made according to a standardized protocols for lymnaeid snails (*Hubendick, 1951*). Shell length was measured for each individual using millimeter paper. Snails were classified into

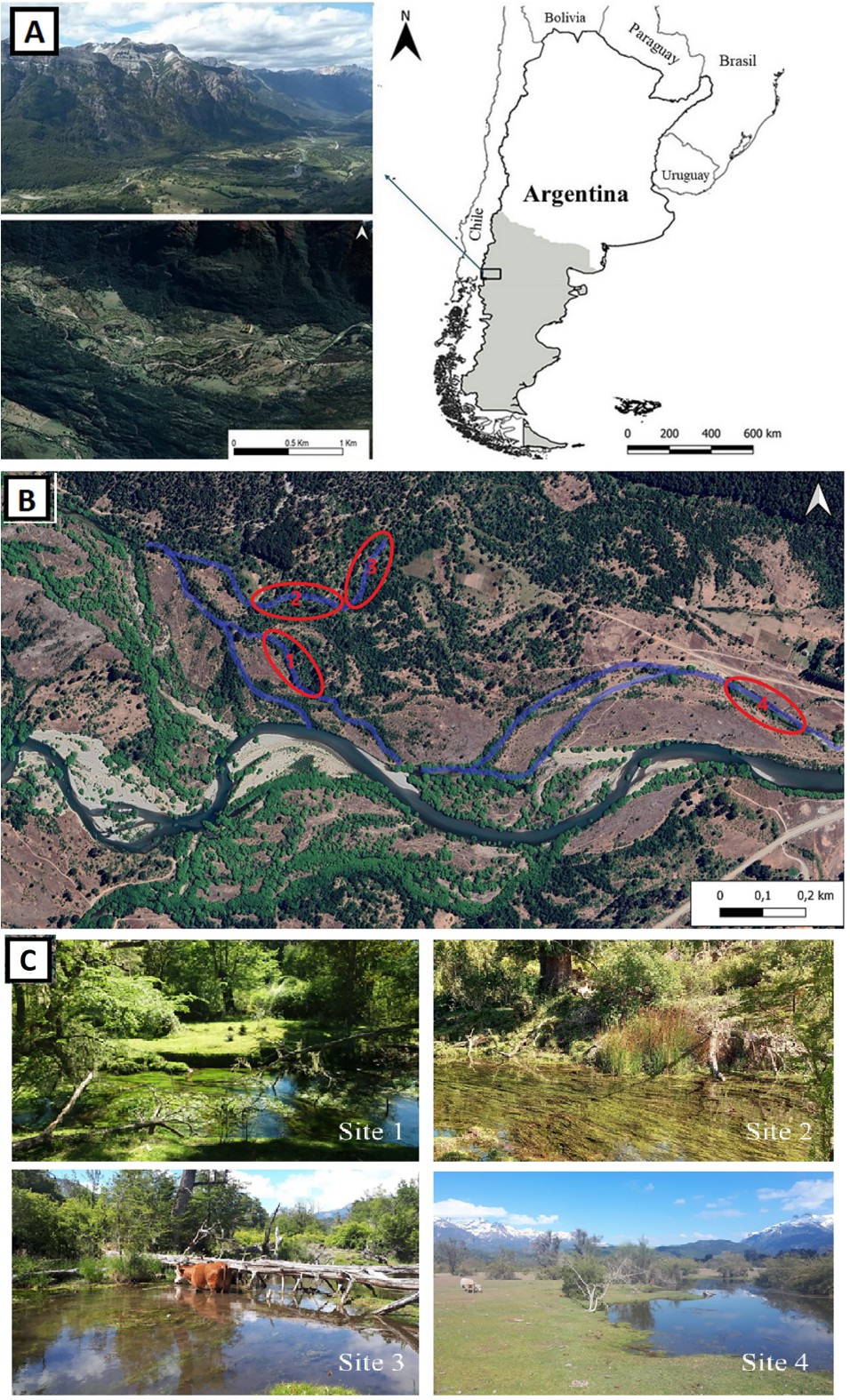

**Figure 1** **(A) Study area, (B) satellite image identifying the sampling sites, (C) four sampling sites.**

three size categories: juvenile (<2.5 mm), pre-adult (2.5–4.5 mm), and adult (>4.5 mm) (*Kendall, 1953*; *Relf et al., 2011*).

### Molecular identification

A total of six individuals were randomly selected for genotyping, and formed part of a larger study of 12 different sites in the provinces of Neuquén, Río Negro, and Chubut (*Soler, Abdala & Larroza, 2023*). Separate PCRs were performed to amplify fragments of the two nuclear internal transcribed spacers (ITS-1 and ITS-2) (*Correa et al., 2010*), and the 18S rRNA gene (*Duffy et al., 2009*). The phylogenetic analysis showed that only *G. viatrix* was found in northern Patagonia (*Soler, Abdala & Larroza, 2023*).

### Assessment of F. hepatica infection

F. hepatica infection in snails was assessed using the crushing technique with clamps on a Petri dish containing water, followed by stereoscopic magnifying glass examination (×40) for the presence of trematode larvae (rediae and cercariae) based on their morphology (*Prepelitchi et al., 2003*; *Relf et al., 2011*). This technique has been used in other studies to evaluate prevalence in lymnaeid populations both in the region and elsewhere (*Kleiman et al., 2007*; *Prepelitchi, 2009*; *Larroza et al., 2014*). Additionally, F. hepatica cercariae were identified to the family level and distinguished from other trematode cercariae found in lymnaeid snails through microscopic examination (×100), based on their taxonomic and morphological characteristics (*Jones, 2005*; *Schell, 1970*).

## Weather data

Data on air and water temperature was registered using a data logger (iButtons/Thermochron) placed in each sampling site. The sensors recorded temperature hourly.

## Statistical modeling

### Population dynamics

Describing (modeling) the relationship between water temperature and snail population dynamics is fundamental to model and understand the ecology of *F. hepatica* transmission. Within our study design, with periodical surveys, a model of the population dynamics of snails should account for: (i) the effects of water temperature on snails, (ii) the population size in the previous survey, and (iii) the possible (and likely) underdetection of snails during surveys as a function of water temperature at the moment of sampling.

Our model consisted of two coupled time series, one for the dynamics of the snail population and another for the observed number of snails, thus separating the ecological process from its observation (*Royle & Dorazio, 2006*; *Gelman & Shalizi, 2013*). The number of collected snails, $y_{t,j,k}$ at survey $t$, in site $j$ and quadrat $k$, was assumed to be a random sample of the "real" (latent) population size (or density), $Y_{t,j}$, under a Beta-binomial distribution with an observation probability $p_{t,j,k}$ and dispersion parameter $\phi$. The logit of $p_{t,j,k}$ was a linear function of water temperature $T^{w}_{t,j,k}$ (measured with a thermometer) of the quadrat $k$ of site $j$ at the time of sampling $t$ with slope $h$ and intercept $g$.

The population dynamics in the model was represented by the variation of the latent variable $Y_{t,j}$. This was assumed to follow a Poisson distribution with rate $\lambda_{t,j}$, which was a linear function of the "real" population size $Y_{t-1_j}$ at the previous survey with the overall population growth rate $\beta_{t,j}$ as slope, and intercept $\gamma$. The rate $\beta_{t,j}$ represented the joint effects of birth, survival, and death, and was modelled as a $T^{\mathrm{m}}_{t,j}$, the average water temperature at site $j$ between surveys $t-1$ and $t$, with slope $\alpha$ and intercept $\delta$.

$$y_{t,j,k} \sim \mathrm{BetaBinomial}\left(p_{t,j,k}, Y_{t,j}, \phi\right)$$

$$\mathrm{logit}\left(p_{t,j,k}\right) = g + hT^{\mathrm{w}}_{t,j,k}$$

$$Y_{t,j} \sim \mathrm{Poisson}\left(\lambda_{t,j}\right)$$

$$\lambda_{t,j} = b_{t,j}Y_{t-1,j} + \gamma$$

$$b_{t,j} = \frac{\kappa_j}{\left(1 + e^{\alpha T^{\mathrm{m}}_{t,j} + \delta}\right)}.$$

A hierarchical structure (random effects) at the site level was considered in $\kappa_j$, the (asymptotic) maximum of the logistic function, representing possible variations in habitat suitability for snail development. All priors were chosen to be weakly informative (see Eq. S1).

### Infection dynamics

We considered that water temperature, snail size, and the previous prevalence of *F. hepatica* could have important effects on the infection probability of snails (*Ollerenshaw & Rowlands, 1959*; *Prepelitchi et al., 2003*; *Bargues et al., 2021*). At each site, we selected average water temperature between surveys as a measure of environmental effects on *F. hepatica* infection in snails. The prevalence of *F. hepatica* in the previous survey (*i.e.*, the fraction of snails found infected in that survey) was used as an indicator of the availability (or exposure) to infectious forms of *F. hepatica* for snails.

The model assumed that the value (0 or 1) of the infection status, $z_{i,j}$, of snail $i$ at site $j$ followed a Bernoulli distribution with infection probability $q_{i,j}$. The logit of this probability was a linear function of water temperature ($T^{\mathrm{m}}_{i,j}$), snail size ($s_{i,j}$), and previous prevalence ($f_{i,j}$), together with their interactions. Site level random effects were included in the intercept $a_j$ to account for possible variations in the suitability for infection at each site (*Bargues et al., 2021*). Priors were chosen to be weakly informative (see Eq. S2).

$$z_{i,j} \sim \mathrm{Bernoulli}\left(q_{i,j}\right)$$

$$\mathrm{logit}\left(q_{i,j}\right) = a_j + b_{\mathrm{s}}s_{i,j} + b_{\mathrm{f}}f_{i,j} + b_{\mathrm{T}}T^{\mathrm{m}}_{i,j} + b_{\mathrm{sf}}s_{i,j}f_{i,j} + b_{\mathrm{sT}}s_{i,j}T^{\mathrm{m}}_{i,j} + b_{\mathrm{fT}}f_{i,j}T^{\mathrm{m}}_{i,j} + b_{\mathrm{sfT}}s_{i,j}f_{i,j}T^{\mathrm{m}}_{i,j}.$$

Models were fitted to data using Markov Chain Monte Carlo methods (*Gelman & Shalizi, 2013*) using JAGS (*Plummer, 2016*) interfaced through R (*R Core Team, 2022*). The population model used five chains, each with 15,000 total iterations, 4,800 burn-in

iterations, and a thinning interval of 3, whereas the infection model used five chains, each with 9,000 iterations, 3,000 burn-in iterations, and a thinning interval of 3. We checked for convergence using r-hat (*Gelman & Shalizi, 2013*). Model validity was assessed using quantile-quantile plots of standardized residuals (as calculated by DHARMa package, *Hartig, 2022*) for each model; very briefly, these Q-Q plots assess how much the values predicted by the model (in our case, the posterior predicted distribution) differ from the observed data. Collinearity was detected between water temperature at the time of sampling and average water temperature between surveys ($r = 0.74$), something to be expected due to the nature of these variables. However, as the fitting to data revealed, the posterior samples of the parameters associated with each of these variables ($h$ and $\alpha$, respectively) showed no significant correlation ($r = 0.02$). In the infection model, the highest correlation was between previous prevalence and temperature ($r = 0.38$). This mild correlation is also to be expected as increasing temperature favors parasite development in snails; for this precise reason, temperature was included as a covariate in the model. Nevertheless, no significant correlation was found between the posterior samples of the respective parameters ($r = -0.04$). In both cases, the uncorrelated posterior samples implies that the estimated parameters were independent between them and, thus, they can be reasonably interpreted each on its own (*McElreath, 2020*).

Field experiments were approved by the National Parks Administration of Argentina (Authorization No. IF-2019-89439453-APN-DRPN#APNAC).

# RESULTS

## Statistical modeling

### Population dynamics

A total of 1,411 *G. viatrix* snails were collected in the 18 periodical samplings along two years. Populations in all four sites tended to show seasonal variation in the number of snails collected, as well as in water temperature. Site 4 showed consistently more snails than the other sites. In all cases, noticeable differences could be observed between seasons as well as sharp variations between surveys adjacent in time. Water temperature showed two distinct dynamics according to the depth of the water bodies. In the deeper sites (1–3), water buffered ambient temperature variations, with 90% of daily means within 8–14 °C (Fig. 2). In contrast, the temperature of the shallower body of water (4) closely followed air temperature, with 90% of daily averages ranging from 2 °C to 21 °C.

We successfully obtained posterior samples from all parameter as all MCMC chains converged (r-hat < 1.1) and the effective sample sizes were >100, except for $\delta$ (N eff = 89) (Table 1). Posterior predictions showed low dispersion and included the observed data in all cases (Fig. S2). Residuals distribution evidenced some underdispersion (Fig. S3), which does not affect model validity; at most, it may imply confidence intervals too wide, something apparently not relevant as all estimated parameters f showed values of virtually 1 (except $\delta$ which was 0.86 nevertheless) (*Harris, Yang & Hardin, 2012*). The model allowed to estimate the posterior distribution of the "true" population abundance (or density), a latent variable, at each time point. The dynamics of this variable provided an

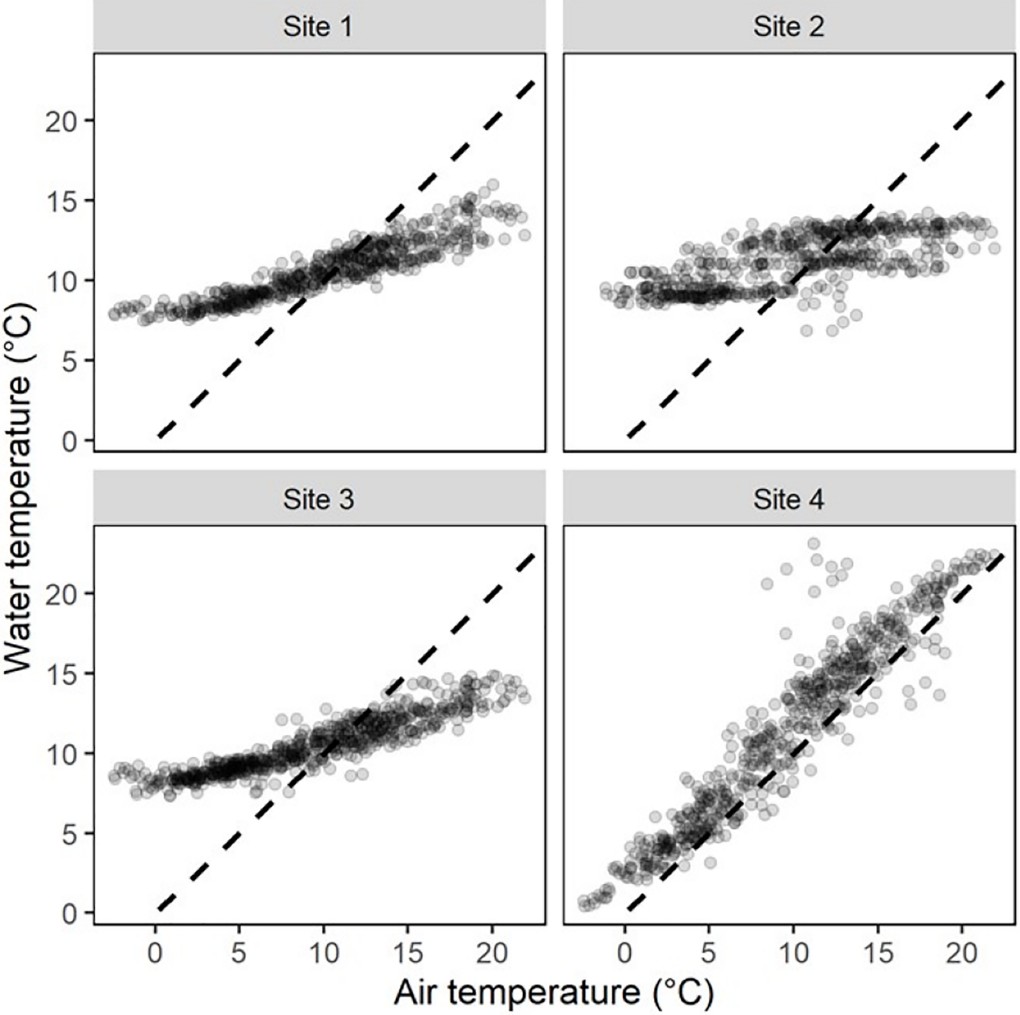

**Figure 2 Relationship between daily mean air temperature and daily mean water temperature at each sampling site.** The broken line indicates identity (*i.e.*, unity slope).

idea of the slightly disjoint dynamics with respect to the observed snail abundance. The "true" abundance tended to increase after the first observed increases in collected snails and to decrease after the observed decreases, consistent with the cumulative effects of temperature on snail development and, eventually, population growth.

Our model implies a specific mathematical relationship of the effects of water temperature. In the case of the effects on the probability of collecting a snail given that it is there, the model showed that water temperature below ~16 °C at the time of sampling translated in almost null probabilities of finding snails (Fig. 3A). In fact, virtually all samplings in sites 1–3 had very low observation probabilities due to low water temperatures at the time of sampling (Fig. 3B). Instead, site 4 showed higher detection probabilities for several surveys as those surveys had water temperatures >20 °C.

Regarding the effects of average water temperature between surveys on population growth rate, the functional relationship changes among water bodies since the maximum

**Table 1 Model fit of the population dynamics of *G. viatrix* snails to water temperature at the time of sampling and mean water temperature between surveys.**

| Parameter | Mean | Q2.5% | Q97.5% | f | R-hat | N eff |
|---|---|---|---|---|---|---|
| $h$ | 0.36 | 0.25 | 0.49 | 1 | 1 | 4,014 |
| $g$ | −10.85 | −14.39 | −7.73 | 1 | 1 | 739 |
| $\varphi$ | 2.45 | 1.29 | 4.21 | 1 | 1.02 | 174 |
| $\mu_\kappa$ | 0.72 | 0.05 | 1.52 | 1 | 1.01 | 427 |
| $\sigma_\kappa$ | 0.62 | 0.21 | 1.31 | 1 | 1 | 1,660 |
| $\alpha$ | 0.28 | 0.11 | 0.46 | 1 | 1.03 | 116 |
| $\delta$ | −0.97 | −2.54 | 1.02 | 0.86 | 1.04 | 89 |
| $\gamma$ | 3.33 | 0.89 | 6.85 | 1 | 1 | 1,166 |

Note:
The mean and 95% interquantile range of the posterior distribution of the corresponding model parameters are shown, together with the fraction *f* of the posterior with the same sign as the mean, r-hat as indicative of convergence and 'N eff', the effective sample size.

of the corresponding logistic functions varied among water bodies. Thus, for sites 1–2, where only low temperatures were registered (<14 °C) the model estimated that the population never grew ($\beta < 1$, Fig. 4); however, site 3, which had similar temperature values, achieved growth rates greater than 1. It should be noted that, exceptionally, at this last site, the water depth was lower during February 2021, when many snails were collected (Fig. S2A). Site 4, with a much wider range of water temperatures, showed winter surveys with $\beta < 1$ and all other surveys with net population growth ($\beta > 1$).

The frequency distribution of snail sizes showed high variability between surveys and among sites (Fig. S4). Large snails (>9 mm) were only observed in sites 3 and 4, and mostly during summer months. In all other months and sites, snail size ranged 3–9 mm. There was no evident seasonal pattern for snails other than large ones.

### Infection dynamics

Overall, infection by *F. hepatica* was detected in 11% of collected snails. Infection was concentrated in 1–2 months per year and only when several snails could be collected (Fig. 5C). Thus, site 3 showed a prevalence of 40% (2/5) in November 2019 and 18% (36/204) in February 2021, whereas site 4 showed prevalence ranging 6–40% (12/197, 65/197, 36/289, and 4/10) in some summer months. No infection was detected in autumn or winter. Prevalence of *F. hepatica* increased steadily with snail size, reaching 32% overall in snails 9–10 mm long (Figs. 5A and 5B). Additionally, other trematode larvae were detected in a total of 60 snails from sites 3 and 4. They were identified at the family level based on the morphological characteristics of the cercariae, as Notocotylidae and Stringeidae which have avian definite hosts. These were present during the same months when we found *F. hepatica* and some times in the same individual snails.

The infection model successfully converged (r-hat < 1.1), with effective sample sizes >1,000 (Table 2). Residuals distribution showed no over or underdispersion (Fig. S3). The model revealed the joint effects of temperature, snail size, and previous prevalence on infection probability. The positive effect of temperature on infection probability increased with snail size and previous prevalence (Fig. 6). Roughly, for snails >6 mm long and with

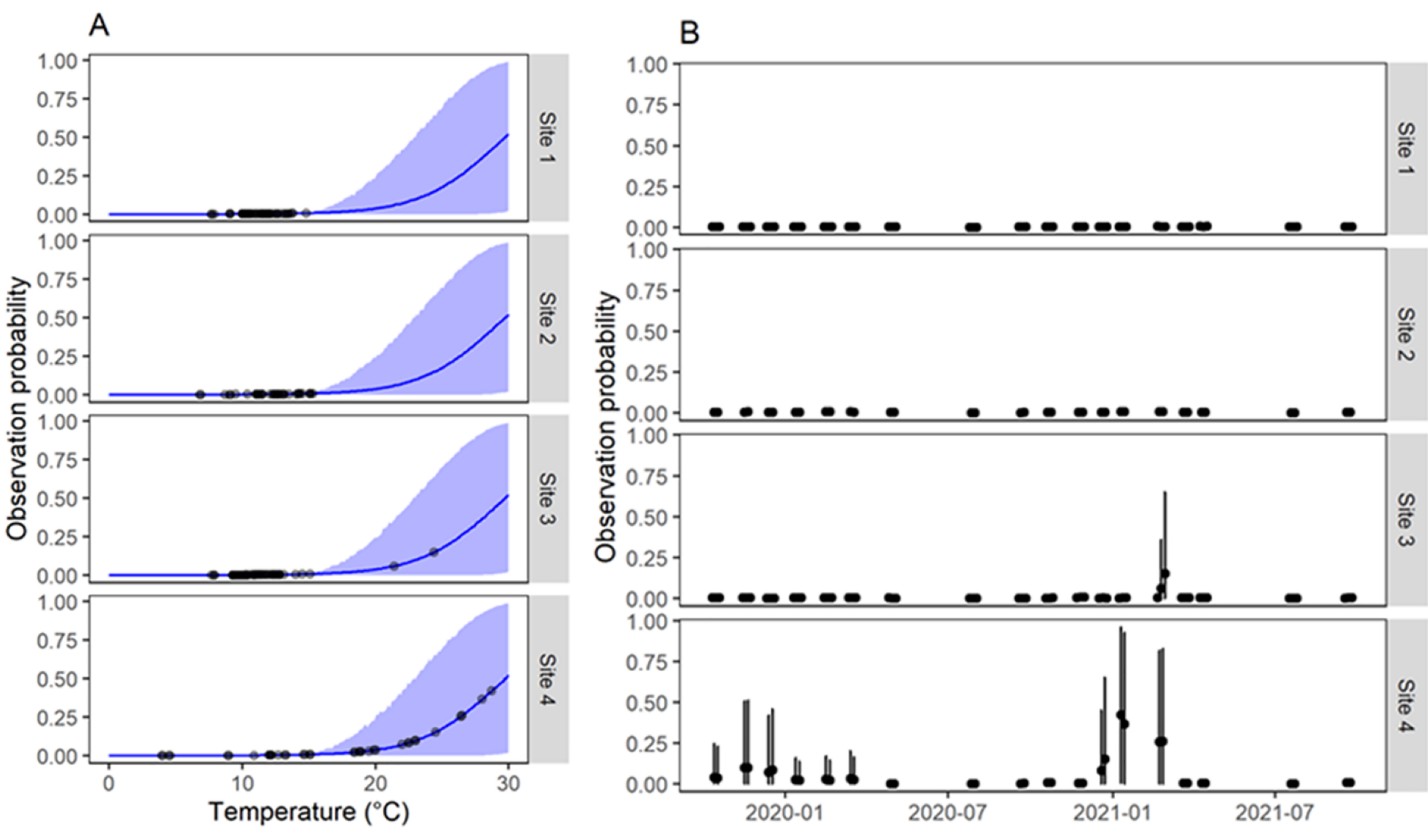

**Figure 3 Estimation of probability of observation of *G. viatrix* snails according to the population dynamics model.** (A) Observation probability as a function of water temperature at the time of sampling. In blue, the mean (line) and 90% interquantile range (shaded area) of the predicted posterior distribution. Black points indicate the mean estimated value for the observed temperatures. (B) Mean (points) and 90% interquantile range (error bars) of the posterior distribution of the observation probability along time for each survey in each quadrant and site.

low previous prevalence, infection probability was very close to zero when water temperatures were <15 °C (Fig. 6A). With low previous prevalence and water temperature >15 °C, only snails >5 mm showed infection probabilities different from zero (Fig. 6B). Previous prevalence strongly shifted the effects of snail size. For instance, with a previous prevalence of 40%, infection probabilities were >0.5 (for sites 1–3) and >0.25 (site 4) even for small snails (<4 mm) at temperatures >15 °C. However, it is noteworthy that very few observations were available for those combinations of values, particularly in sites 1–3. In fact, the uncertainty in the estimated infection probability was greatest for high previous prevalence and for sites 1 and 2 (those with fewer collected snails).

## DISCUSSION

Using high-frequency data (monthly for snail monitoring and hourly for water temperature), we were able to statistically model snail population growth rate, probability of detection, and probability of infection by *F. hepatica* in snails. Our data show high variability inherent to this system, both between dates (even within the same season and site) and between sites and seasons. Although the four sampling sites were in the same area and therefore exposed to the same air temperature, water temperature at each site

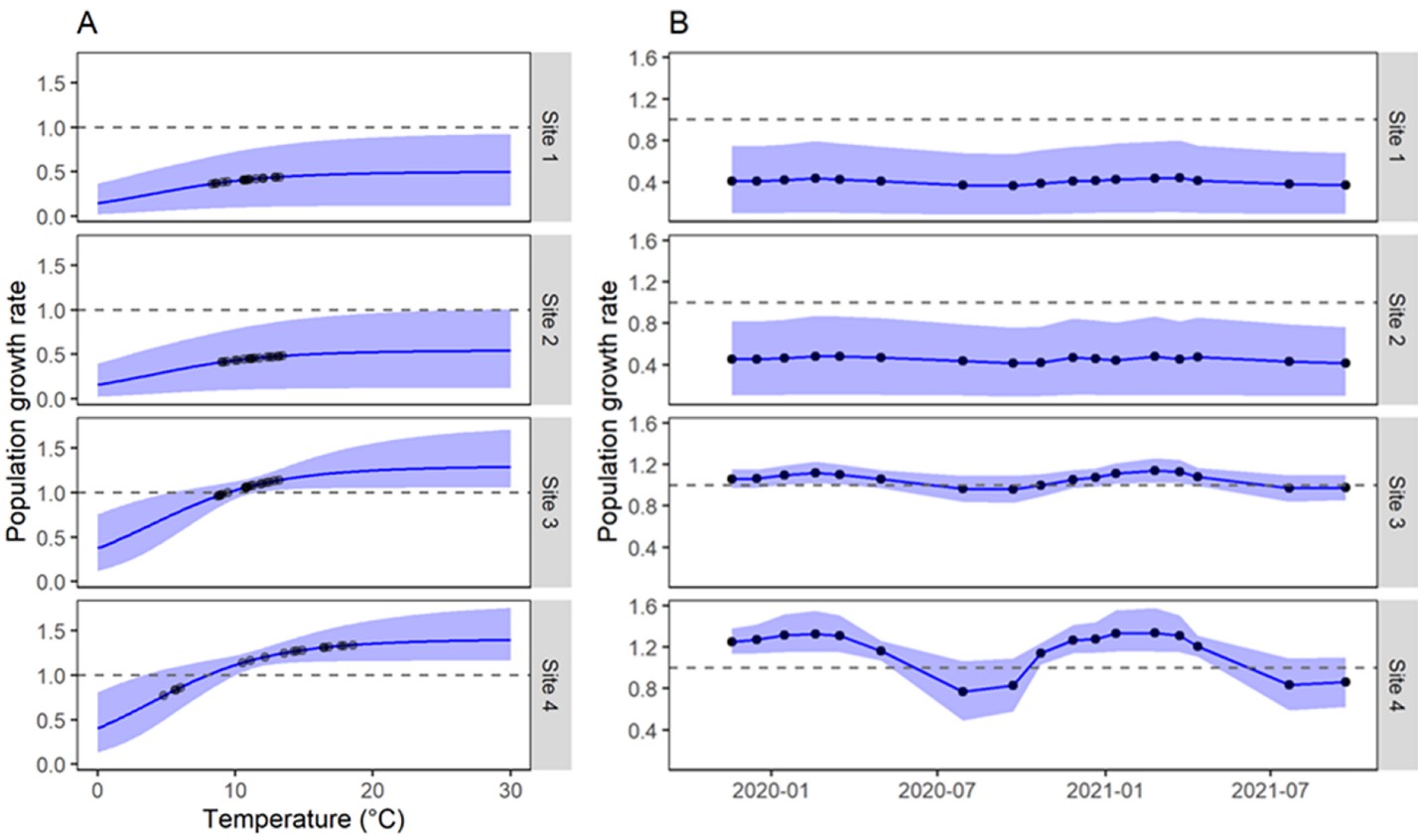

**Figure 4 Population growth rate of *G. viatrix* snails according to the population dynamics model.** (A) Growth rate as a function of mean water temperature between surveys as estimated from the joint posterior distribution of the fitted model. (B) Estimated growth rate for each survey and site. Black points indicate the values estimated for the actual surveys. The blue line and shaded area display the mean and 90% interquantile range of the posterior distribution as estimated by the model. A growth rate >1 implies a net population growth, whereas values <1 denote a net population decrease.

exhibited different behaviors depending on the depth of the water body. This translated into variability in detection and abundance of snails at each site.

Previous studies with have suggested that lymneids prefer shallow sites because they require access to the surface to breathe (*e.g.*, *Ruppert & Barnes, 1996*). Here, we suggest that water depth is also important because it affects temperature and thus snail development. Indeed, at sites 1–3, with greater depths (0.25 to 0.5 m), the buffering of air temperature variations was higher, resulting in water temperatures being confined within a narrower range throughout the year (monthly mean temperatures of 8–14 °C). This differs from what was observed at site 4, which was very shallow (<0.1 m). There, water temperature tended to be similar to air temperature, with pronounced seasonal fluctuations, reaching much more extreme temperatures than those observed at the other sampled sites (monthly mean temperatures ranging from 2 °C to 21 °C). According to our model, the consistently more abundant population of *G. viatrix* at site 4 can be attributed to the higher water temperatures compared to the other sites. In fact, 92.4% of the total collected snails were found at site 4, and the highest prevalences of *F. hepatica* in snails were also found at this site. This suggests that, in this study region, the increase in water temperature, associated

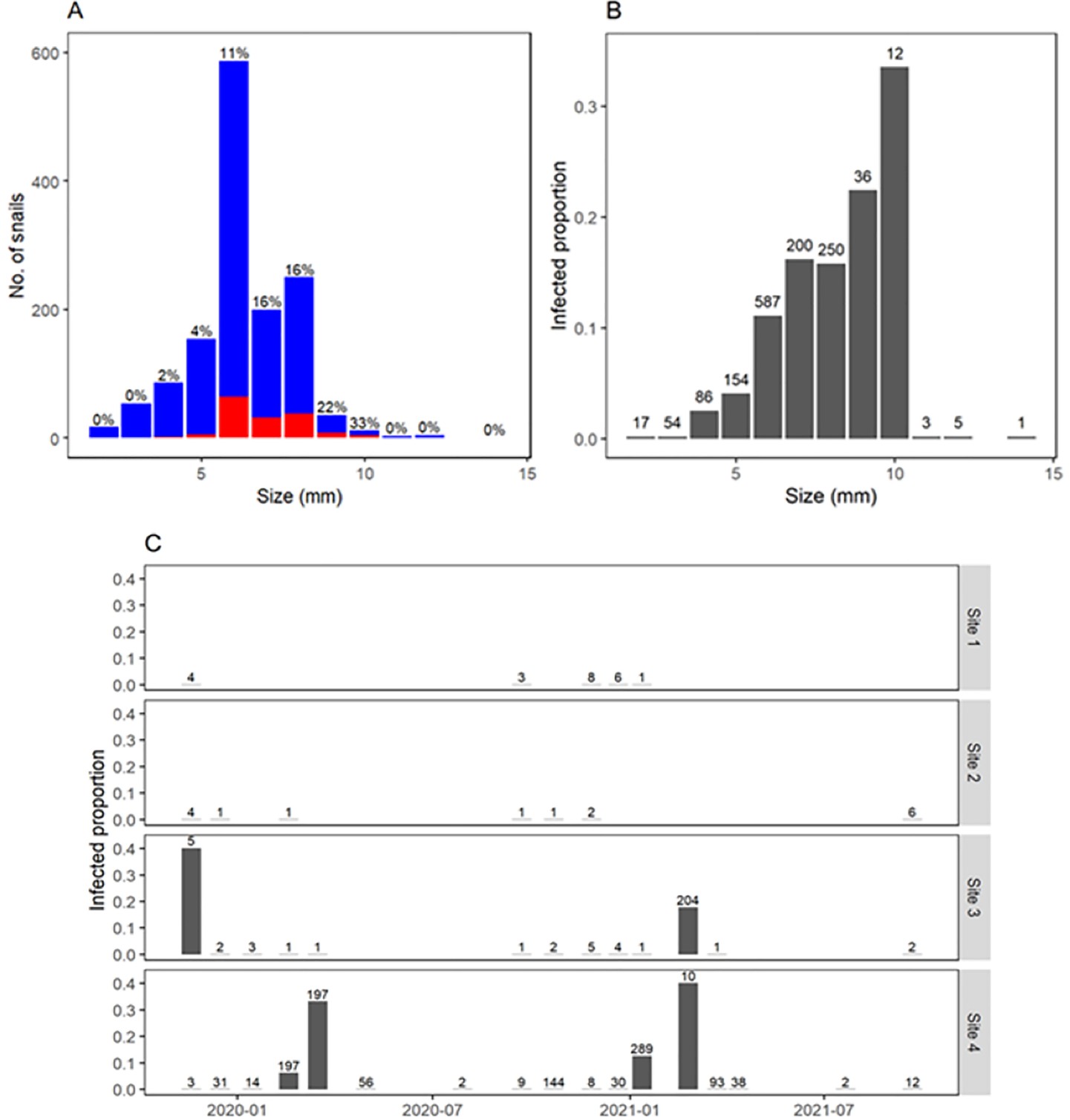

**Figure 5** *Fasciola hepatica* **infection and body size of *Galba viatrix* snails.** (A) Number of collected snails according to snail size (rounded to the nearest millimeter) and infection (red, infected; blue, uninfected). (B) Relative frequency of infected snails according to their body size. (C) *F. hepatica* prevalence in each survey and site along time. The total number of collected snails (infected and uninfected) is indicated above bars.                                                    

**Table 2 Model fit of the *F. hepatica* infection dynamics to water temperature, snail size, and *F. hepatica* prevalence in each previous survey.**

| Parameter | Mean | Q2.5% | Q97.5% | $f$ | r-hat | N eff |
|---|---|---|---|---|---|---|
| $b_{size}$ | −0.70 | −1.10 | −0.33 | 1.00 | 1.00 | 1,029 |
| $b_{prev}$ | 0.10 | −0.77 | 0.97 | 0.58 | 1.00 | 3,670 |
| $b_{temp}$ | −0.19 | −0.32 | −0.05 | 1.00 | 1.00 | 3,763 |
| $b_{size \times prev}$ | 0.34 | −0.46 | 1.14 | 0.80 | 1.00 | 10,000 |
| $b_{size \times temp}$ | 0.07 | 0.04 | 0.1 | 1.00 | 1.00 | 1,403 |
| $b_{prev \times temp}$ | 0.72 | 0.27 | 1.17 | 1.00 | 1.00 | 6,771 |
| $b_{size \times prev \times temp}$ | −0.04 | −0.12 | 0.04 | 0.86 | 1.00 | 10,000 |
| $a_1$ | −3.27 | −5.03 | −1.55 | 1.00 | 1.00 | 6,938 |
| $a_2$ | −3.19 | −5.04 | −1.42 | 1.00 | 1.00 | 2,474 |
| $a_3$ | −2.15 | −3.38 | −0.86 | 1.00 | 1.00 | 2,503 |
| $a_4$ | −3.85 | −5.22 | −2.44 | 1.00 | 1.00 | 5,856 |

Note:

The mean and 95% interquantile range of the posterior distribution of the corresponding model parameters are shown, together with the fraction *f* of the posterior with the same sign as the mean, r-hat as indicative of convergence and 'N eff', the effective sample size.

with the shallow depth of the water bodies, facilitates the development of snails and *F. hepatica* (*Rodriguez Quinteros et al., 2024*). Similar findings by *Bargues et al. (2021)* for *G. truncatula* highlight water temperature as a crucial factor. Variations in water temperature across sites impact the abundance and dynamics of lymnaeid populations, with snail colonies showing less growth in deeper waters compared to shallow sites.

The model allowed us to estimate a curve depicting the effects of water temperature on the probability of snail detection. Under the conditions of this study, there would be very low chances of finding snails below ~16 °C. As the temperature rises above 16 °C, the detection probability tends to increase, albeit with increasing uncertainty, reaching values close to 1 at temperatures near 30 °C. Only this type of hierarchical models, with latent variables, makes it possible to separately estimate the dynamics of population abundance and the observation process. Both processes depended on water temperature but operated at different temporal scales. Our results also show that temperature at the time of sampling can strongly distort the estimation of population abundance: during summer, with high monthly mean temperatures, if the temperature at the time of sampling was low (*e.g.*, <20 °C), only a few snails were captured, even if the population at the site was abundant (as demonstrated by samplings in adjacent months). This agrees with what has been found by *Rodriguez Quinteros et al. (2024)* for this same species and region.

According to experimental studies, snail development does not occur below 10 °C (*Ollerenshaw, 1971*; *Aziz & Raut, 1996*; *Claxton et al., 1999*). This is consistent with field studies that found a monthly mean air temperature of 10 °C as limiting for snail activity (*Boray, 1964*; *Ollerenshaw, 1971*; *Malone et al., 1984*; *Claxton et al., 1999*; *Olaechea, 2007*). However, in this study, we assessed the effects of water temperature, rather than air temperature, on lymnaeid populations, as snails are in contact with water rather than air. In fact, our results showed active snails during spring (October–November), when the

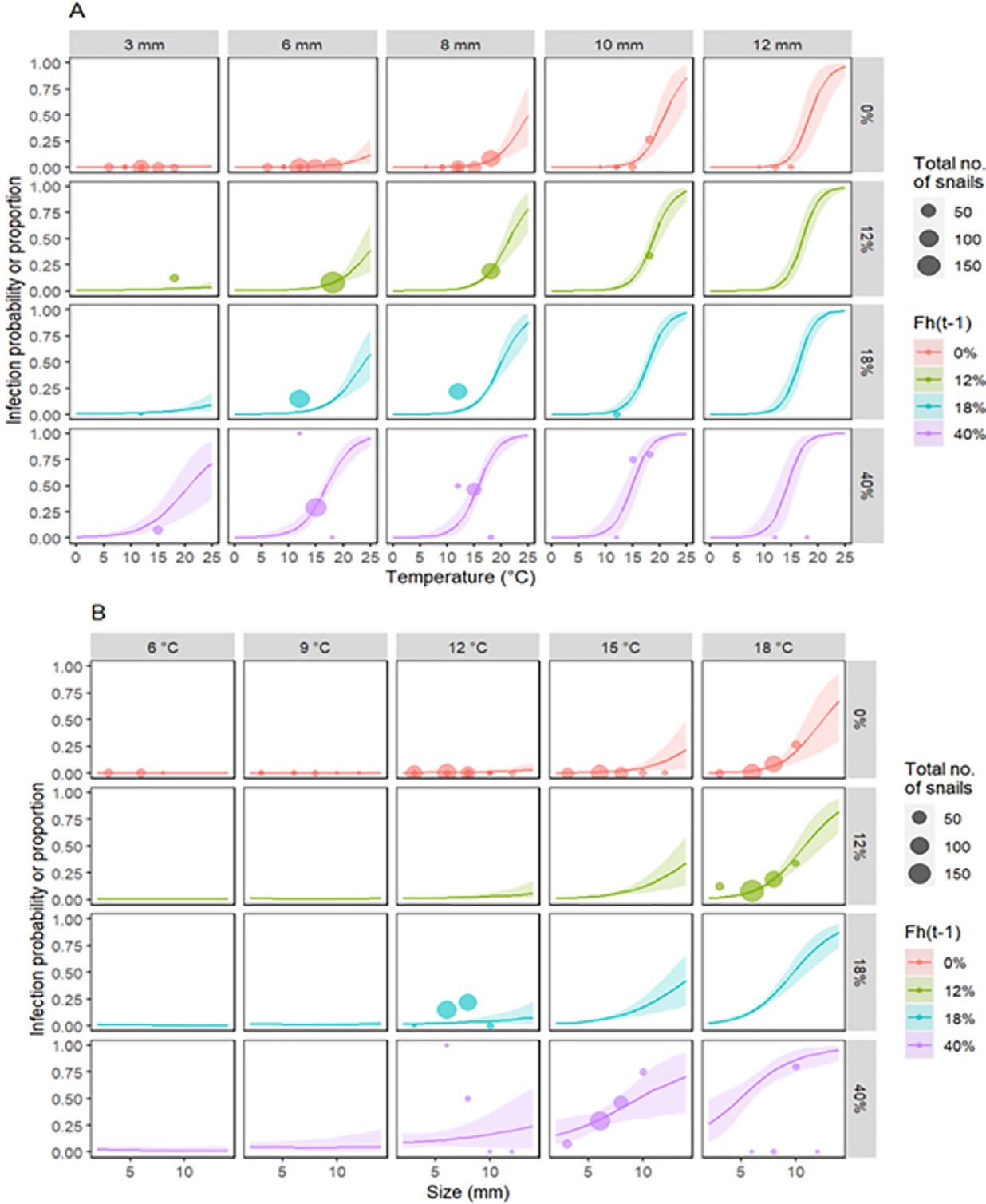

**Figure 6 Site 4: Predictions of the infection probability according to the fitted model of *F. hepatica* infection in snails as a function of their size, water temperature, and *F. hepatica* prevalence in each previous survey.** Lines and shaded areas represent the model predictions (mean and 90% interquantile range, respectively). Colors indicate different values of previous prevalence. Points indicate the observed values. The size of points represents the total number of snails assessed. (A) Infection as a function of temperature for different snail sizes (columns). (B) Infection as a function of snail size for different water temperatures (columns).

monthly mean air temperature was below 10 °C, but the monthly mean water temperature was above this value (12–15 °C). Our results provide evidence, under field conditions, of a threshold close to 10 °C for water temperature. However, precisely due to these field conditions and the use of a site-specific hierarchical model, we observed that this value may vary among sites, with some sites where no temperature guarantees a net population growth. This suggests that specific characteristics of each site related to how favorable the site is (beyond water temperature) for snail development also determine the population dynamics of snails. On the other hand, references regarding the 10 °C limit for the presence of lymnaeid snails have generally focused on studies of *G. truncatula* (*Ollerenshaw, 1971*; *Malone et al., 1984*); however, few studies have focused on *G. viatrix*. Considering that molecular analyses have revealed that the dominant species in the study area is *G. viatrix* (*Soler, Abdala & Larroza, 2023*), it is important to understand what its specific temperature preferences are and how the dynamics of these snails vary compared to more widely studied genera.

The hierarchical model allowed to estimate the "true" snail abundance by using a latent variable and considering a variable detection probability. In sites where abundant snails were found one or more times (sites 3 and 4), the "true" abundance showed increasing uncertainty, including higher abundances towards the end of our time series. As it was the last portion of the time series, there were no data on future abundances. Additionally, the last months correspond to winter, with expected lower "true" abundances, but even more so, with very low detectability and therefore greater uncertainty in the estimation of "true" abundance. Consequently, since the model does not consider any function that imposes periodicity, it is expected that the model will achieve lower precision in the final segments of the time series. In contrast, throughout the previous parts of the time series, the model successfully captures seasonality in sites 3 and 4, as shown by the variations in "true" abundance and the estimated population growth rate. All of this highlights the importance of having long time series, spanning more than two years, to properly characterize these cycles.

The observed seasonality of snail abundance coincides with previous studies in different countries (*Kendall & Ollerenshaw, 1963*; *Malone et al., 1984*; *Torgerson & Claxton, 1999*; *Prepelitchi, 2009*; *Bargues et al., 2021*). The details of the seasonality may be specific to the study area and possibly the region (*Rubel et al., 2005*; *Kleiman et al., 2007*; *Olaechea, 2007*; *Larroza et al., 2014*). Nevertheless, the relationship with temperature could be extrapolated to bodies of water in other areas, at least regarding *G. viatrix*. It is interesting to speculate that the depth and, therefore, the degree of variability in water temperature will have a different effect depending on the range of air temperature values.

Previous reports in Argentina show the wide range of values that the prevalence of *F. hepatica* in *G. viatrix* snails can reach in the field, with prevalences of up to 14% recorded during the summer months (*Rubel et al., 2005*; *Kleiman et al., 2007*; *Larroza et al., 2014*; *Vazquez et al., 2023*). In this study, we found a maximum prevalence of 40% (2/5 in November 2019 and 4/10 in February 2021), followed by 33% (65/197 in March 2020). These are the highest prevalences recorded in *G. viatrix* populations naturally infected with *F. hepatica* in Argentina to date. In fact, these prevalences can be compared with those

reported in Peru (48%) for *G. viatrix* (*Londoñe et al., 2009*), as well as with prevalences reported in other species such as *G. bulimoides* (77%), and *G. humilis* (33%) in Mexico (*Cruz-Mendoza, Figueroa & Correa, 2004*; *Cruz-Mendoza, Ibarra & Quintero, 2005*), *G. cousini* (46%) in Ecuador (*Celi-Erazo et al., 2020*), *Pseudosuccinea columella* (41%) in Peru (*Londoñe et al., 2009*), and in large grazing areas in several countries in Europe and Africa where *G. truncatula* is responsible for transmission (*Vazquez et al., 2023*). These high values could be attributed to the presence of favorable environmental conditions for the development of lymnaeid snail populations in areas endemic to fasciolosis, such as perennial watercourses and continuous sheep grazing throughout the year. Indeed, these definitive hosts are considered to be the primary contributors to the ongoing contamination of pastures in Patagonia (*Olaechea, 1994*). Current climate change has affected the seasonality and abundance of snails, as well as directly influenced the reproduction rates of the parasites they carry (*Mas-Coma, Valero & Bargues, 2009*). This could be occurring with lymnaeid populations in northern Patagonia, where there is evidence of a recent increase in temperature (*Barros & Vera, 2014*; *Pessacg et al., 2020*), which could favor an increase in local prevalence and intensity and an expansion of their geographical distribution (*Seeber et al., 2024*).

Considering the high prevalence found in snails and the high density detected per square meter (288 snails in January 2021 in site 4), it would be expected that there is a high risk of parasite transmission to animals in the studied area (*Ollerenshaw & Rowlands, 1959*). Our model for *F. hepatica* infection in snails found important effects of temperature, snail size, and previous infection in the site but we do not know what was the availability of contaminated faeces from hosts in the environment. Also, we found strong seasonality in population dynamics of snails and their infection. Some studies (*Parr & Gray, 2000*; *Jones et al., 2017*) have found an association between fluke infection levels in snails an in grazing livestock. However, we speculate that infection risk in our system would persist throughout the year due to the ability of metacercariae to remain infective for several months in the environment (*Amato et al., 1986*; *Olaechea, 2007*). Further studies should consider host's seasonal space use in order to evaluate the importance of this component in the seasonality of the *F. hepatica* life cycle.

Consistent with previous reports in the literature (*Kendall, 1953*; *Relf et al., 2011*), no snails <4.5 mm long were found to be infected with *F. hepatica*. Above that size, the prevalence increased as the size increased. Interestingly, the highest prevalences (>22%) were found in the less common snail sizes (≥9 mm), which were more abundant towards the end of summer. These snail sizes are relevant in terms of environmental contamination with *F. hepatica* in autumn and serve as a biological reservoir for the subsequent parasite population expansion in spring (*Pruzzo, 2019*).

Regarding the other trematode larvae found in the analyzed snails, they were identified as belonging to the families Notocotylidae and Stringeidae, which have avian species as their definitive hosts. It is important to highlight that these findings occurred solely during the summer. This suggests that, similar to what was observed with *F. hepatica*, the infection of snails by other trematodes is also directly related to the increase in water temperature and the availability of large snails.

Although our research provides relevant information about the population dynamics of *G. viatrix* snails in the studied region, it is important to note some limitations. Firstly, the small number of sampled sites, among which only one showed a large snail population. Additionally, there are challenges in making inferences about seasonality and variability among years and sites, as the sampling was conducted for only two years. Another limitation is the exclusive focus on water temperature as an environmental variable. While temperature is a key factor for the development and survival of snails (*Kleiman et al., 2007*), other factors such as pH, water depth, soil type, vegetation, and water flow could influence population dynamics (*Boray, 1964*, *1969*; *Yigezu et al., 2018*; *Bargues et al., 2021*; *Roessler et al., 2022*). However, since several of these variables tend to remain relatively constant within a single site, a considerable number of sites would be required to achieve adequate statistical power. The significant effort needed to conduct more extensive sampling, both in terms of time and the number of sites, is essential for discerning the underlying processes in such a variable system.

## CONCLUSION

Overall, our results highlight the significant temporal and spatial variability in the population dynamics of snails and the prevalence of *F. hepatica* in them. Our model quantified the effects of water temperature on the detection probability and population growth of *G. viatrix*. This is a crucial input for generating mechanistic models of *F. hepatica* transmission that would facilitate the design and simulation of potential interventions based on improved treatments and on environmental and livestock management, taking into account the specific characteristics of each region.

## ACKNOWLEDGEMENTS

The authors wish to acknowledge the assistance of Tech. Raúl Cabrera for his collaboration in field sampling.

### Funding

This work was supported by Agencia Nacional de Promoción la Investigación, el Desarrollo Tecnológico y la Innovación (the project PICT 2017-0787) and by the Instituto Nacional de Tecnología Agropecuaria (The project PD-I115 INTA). The funders had no role in study design, data collection and analysis, decision to publish, or preparation of the manuscript.

### Grant Disclosures

The following grant information was disclosed by the authors:
Agencia Nacional de Promoción la Investigación, el Desarrollo Tecnológico y la Innovación: PICT 2017-0787.
Instituto Nacional de Tecnología Agropecuaria: PD-I115 INTA.

## Competing Interests

The authors declare that they have no competing interests.

## Author Contributions

- Paula Soler conceived and designed the experiments, performed the experiments, analyzed the data, prepared figures and/or tables, authored or reviewed drafts of the article, and approved the final draft.
- Juan Manuel Gurevitz conceived and designed the experiments, analyzed the data, prepared figures and/or tables, authored or reviewed drafts of the article, and approved the final draft.
- Juan Manuel Morales conceived and designed the experiments, analyzed the data, authored or reviewed drafts of the article, and approved the final draft.
- Marcela Larroza conceived and designed the experiments, performed the experiments, authored or reviewed drafts of the article, and approved the final draft.

## Animal Ethics

The following information was supplied relating to ethical approvals (*i.e.*, approving body and any reference numbers):

All procedures related to the sampling and handling of snails from protected areas were approved by the National Parks Administration of Argentina (Authorization No. IF-2019-89439453-APN-DRPN#APNAC). We adhered to applicable national guidelines for the care and use of animals during all procedures conducted.

## Field Study Permissions

The following information was supplied relating to field study approvals (*i.e.*, approving body and any reference numbers):

Field experiments were approved by the National Parks Administration of Argentina (Authorization No. IF-2019-89439453-APN-DRPN#APNAC).

## Data Availability

The raw data is available in the Supplemental Files.

## Supplemental Information

Supplemental information for this article can be found online at http://dx.doi.org/10.7717/peerj.18648#supplemental-information.

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
