# Peer review of "Modeling the effects of water temperature on the population dynamics of Galba viatrix and infection by Fasciola hepatica: a two-year survey in Andean Patagonia, Argentina"

_PeerJ, doi:10.7717/peerj.18648_

## Round 0.1 · original submission · Major Revisions

Could you please address reviewers' concerns, particularly regarding the determination of the snails' infection status and details about the study sites? In addition, some figures require improvement.

Reviewer 1 ·

Basic reporting

The clarity of the English language could be enhanced in certain sections, for example, in lines 110-112 or most of the discussion part. Refining the phrasing in these areas would improve reader comprehension and strengthen the overall presentation of the research.


- Please consider rearranging the photos in Figure 1 so that the north arrow appears accurately.
- Figure 6 and Table 2 seem to be absent from the text. Please ensure they are referenced where appropriate.

Experimental design

Line 134: Consider adding information about the time of day when sampling was conducted.
Line 134: It seems that the "quadrant" method is mentioned throughout the article; however, the correct term might be "quadrat." Please review and revise accordingly.
Line 169: Could you specify which larval stage is being referred to?

Validity of the findings

Line 376: I suggest discussing the reasons behind the observed increase. What is your perspective on the potential impact of climate change?

Additional comments

Dear Authors,

Thank you for your valuable research and for contributing to our understanding of snail dynamics. This is a well-conducted study, and I have a few suggestions that may help improve the manuscript:

Line 55: Please provide a citation for the presence of "F. hepatica" in the Americas.
Line 61: Please remove the last semicolon after 2023.
Line 69: A citation is needed to support the mention of 30 species.
Line 74: There is a missing space before the word "in northern"
Line 95: Ensure there is a space between the "comma" and "1959"
Line 225: There should be a space between "probability" and "was."
Line 384: Please italicize F. hepatica.

Reviewer 2 ·

Basic reporting

Overall reporting is ok with space for some improvements.
1. More context needed when discussing previous research in the field to clarify which Galba species specific facts are proven for.
2. Improvements needed to figures including axis text size, figure titles and resolution of points on figures.

Experimental design

Areas that need improvement here:
1. More detail on study sites and clarity regarding the sampling and analysis process, specifically how quadrant location within sites were selected and how F. hep infection status of snails was determined.
2. Validate model and report results of model residual analysis, ICC of model etc.

Validity of the findings

Very interesting finding regarding relationship with water/air temperature. Some questions regarding validity of findings re fluke infection status which needs clarifying.

Additional comments

I particularly enjoyed reading about water temperature and how it varies from air temperature in these areas. When modelling liver fluke, air temperature is almost always used within models, but clearly water temperature may be more suitable and thus these findings will build into a body of evidence that can guide future fluke forecasting efforts.

Nevertheless, there are some issues present with this paper and these must be addressed or clarified if I have been mistaken before publication.

Firstly, I have doubts regarding the modelling of fluke infection levels aspect and would ask that the manuscript is revised to add more clarity on the methodologies for this aspect. Mainly there is a lack of detail in regard to how F. hep infection status was determined. Which life stages were identified and how were these differentiated from other trematode species? Which other trematode species were detected and was the predominant trematode infection variable between habitats and time? There is also an opportunity to analyse or at least add some descriptive information regarding the seasonality of the fluke lifecycle. Was there a time when cercariae appeared and was this associated with water temperature/other variables? This detail would help to add context to the seasonality of fluke infection in snails in this region.

There are also questions in regard to the utility of water temperature as a predictor or determinant of fluke infection status in snails. Water temperature will determine the abundance and size and survival of snails, and in effect that will lead to the number of susceptible snails available for fluke infection and development. But the degree of eggs/miracidia in the environment will also be a key factor. I realise that this data was not collected (i.e egg count of grazing animals or in faecal material in the vicinity) and that previous snail infection was used as a proxy for this in the models, but there should be some discussion of this including a description of livestock/wildlife in the area, their history (treatments, infections etc.) and the periods where they accessed the sampled habitats. Temperature will also influence egg hatching, so there should be discussions of this key life stage as well.

Introduction
This is mostly fine, but I do feel that it could be improved by focusing more on G. viatrix. For example many statements are made regarding snail preferences with references used in most instances studies that focused on G. truncatula. For example you highlight that snail activity is triggered by temp over 10*C, but this information is specific to Galba truncatula (Ollerenshaw + Malone). I think a key point to highlight here is that as molecular analysis reveals that other Galba species are predominant in certain areas i.e as we see in the study area, we need to understand how these specific snails dynamics and preferences vary from the more widely studied G. truncatula. E.g do we know for certain that G. viatrix also becomes active at 10*C, has anyone looked at this? I would advise that this is discussed in the introduction, with a point made that there is limited studies that have specifically focused on G. viatrix. This would also help justify your study.
Methods
Line ~125 – more detail needed for each study site/ Perhaps a table could be included with key information regarding each. Specifically lacking is detail on site 4, what type of habitat was this? Could also include photographs of each location.
Line 131 – aestivate rather than hibernate?
134 – searched for
Line 136 – why 2-3 quadrants? Why no consistency i.e always 2 or always 3? Also how were quadrants locations selected? I ask as snail presence within a habitat may not be evenly distributed across the site. Were there any steps to minimise risk of bias, for example what ensured random selection of quadrant location. This is particularly important as you conclude that water depth was a key factor on determining snail abundance (indirectly).
Line 150 – was each study site represented by the 6 sequenced snails? Did they range across time periods/sizes, infected . uninfected? Please give details.
167 – I assume there was no molecular ID of F. hepatica infection. Therefore, please provide detail of how F. hepatica infection was confirmed. Were there specific stages that were identified (sporocysts, rediae, cercariae) and what features were considered to confirm identification? Were any other trematode infection identified? This section is also slightly confusing as you note snails were dissected and then twice repeat that they were crushed. Please clarify and consider re wording this section.
Section 2.4
Although detailed information is given regarding model building, there is a lack of information regarding how the validity of the model was assessed. Were model diagnostics tests employed? Were model residuals assessed for their distribution pattern, the presence of outliers or zero inflation. Was there any collinearity between variables in the model and as there were random effects in each model, we need to understand the degree of variation these random effects caused in each model, perhaps using an ICC value. If this ICC is substantial further discussion of reasons for variation between sites should be discussed.
Results
Sub sections are advised here (descriptive – snail abundance – F. hep infection)
Detail of F. hep infective stages + other trematode infections would also be useful
Discussion
Line 316 – this is a valid hypothesis you present, and I do agree with it to some extent. However, we do know that Galba snails have a general preference for shallower water with some hypothesising that this is due to the fact that they are air breathing and require access to the surface to breathe. This point is worthy of inclusion, although you work does provide evidence that temperature and not access to surface may drive this preference which is interesting.
Line 340 – as noted in the intro it is worth clarifying differences between species for preferences to temperatures and discussing this here. I.e does your results suggest that G. viatrix has similar temperature preferences compared to other galba species.
Line 384 – I’m not sure if the findings of Amato (1986) are consistently seen in other studies. In fact there are a number of studies (Parr and Gray, 2000; Jones et al., 2017 etc.) that have shown strong associations between fluke infection levels in snails an in grazing livestock. This also highlights differences in fluke seasonality in livestock v. snail with heavy livestock risk in winter and snails in summer.
Line 397 – examples are given of other factors other than water temperature that may influence. Similarly, you could expand to note that other factors are likely to influence fluke infection levels. None more so than the degree of fluke egg contamination on habitats.

Figures and supplementary content
Figure 1 – we need to see more clarity in the figure title. What do the two blue lines represent, separate watercourses? Why are there 4 quadrants on the figure, yet on the title it says 2-3 quadrants were selected at each sites? What is the black circle – the sampling area? As there was only four sampling sites perhaps a labelled satellite image or photograph of each would be more suitable?

Figure 2A and Figure 3 – very hard to read figures and I struggle to see the red points (Fig 2) and Orange points (fig 3) only the error bars. Size category information on X axis in figure 3 is far too small and cannot be read

Supp file 1 – there are 2 qty_quad_2 columns

---

## Round 0.2 · Minor Revisions

Please make recommend minor changes by the reviewer

Reviewer 2 ·

Basic reporting

no further comment

Experimental design

no further comment

Validity of the findings

no further comment

Additional comments

I congratulate the authors on the improvements made to this manuscript. I am now happy, following the changes made, to advise that the paper is to be accepted subject to a small number of minor amendments.

1. Line 272 - “but the process we modeled inherently implied dealing with both temperature variables.” – please re word this sentence as it is currently unclear what the quoted part of the sentence means. Perhaps split the sentence into two, and specifically explain (with references) why a high collinearity value in your model is not an issue.

Line 297 – “Residuals distribution evidenced some underdispersion, which does not affect model validity (Fig. S3).” With the QQ plots now presented in figure S3 readers can assess the residuals of the models which is a very important improvement, however, I would like for you to elaborate on why the underdispersion does not affect model validity, ideally with references to back up reasons. This would alleviate any concerns that a reader may have when seeing underdispersion in the supp figure,

References - The Jones et al., 2017 reference suggested during the review process should be cited as the following rather than the paper you have cited:

Jones, R.A., Williams, H.W., Dalesman, S., Ayodeji, S., Thomas, R.K. and Brophy, P.M., 2017. The prevalence and development of digenean parasites within their intermediate snail host, Galba truncatula, in a geographic area where the presence of Calicophoron daubneyi has recently been confirmed. Veterinary Parasitology, 240, pp.68-74.

---

## Round 0.3 · accepted · Accept

The authors have successful addressed the concerns of all reviewers